# Bandits with Unobserved Confounders:
# A Causal Approach

**Elias Bareinboim**\*
Department of Computer Science
Purdue University
eb@purdue.edu

**Andrew Forney**\*
Department of Computer Science
University of California, Los Angeles
forns@cs.ucla.edu

**Judea Pearl**
Department of Computer Science
University of California, Los Angeles
judea@cs.ucla.edu

## Abstract

The Multi-Armed Bandit problem constitutes an archetypal setting for sequential decision-making, permeating multiple domains including engineering, business, and medicine. One of the hallmarks of a bandit setting is the agent's capacity to explore its environment through active intervention, which contrasts with the ability to collect passive data by estimating associational relationships between actions and payouts. The existence of unobserved confounders, namely unmeasured variables affecting both the action and the outcome variables, implies that these two data-collection modes will in general not coincide. In this paper, we show that formalizing this distinction has conceptual and algorithmic implications to the bandit setting. The current generation of bandit algorithms implicitly try to maximize rewards based on estimation of the experimental distribution, which we show is not always the best strategy to pursue. Indeed, to achieve low regret in certain realistic classes of bandit problems (namely, in the face of unobserved confounders), both experimental and observational quantities are required by the rational agent. After this realization, we propose an optimization metric (employing both experimental and observational distributions) that bandit agents should pursue, and illustrate its benefits over traditional algorithms.

## 1 Introduction

The Multi-Armed Bandit (MAB) problem is one of the most popular settings encountered in the sequential decision-making literature [Rob52, LR85, EDMM06, Sco10, BCB12] with applications across multiple disciplines. The main challenge in a prototypical bandit instance is to determine a sequence of actions that maximizes payouts given that each arm's reward distribution is initially unknown to the agent. Accordingly, the problem revolves around determining the best strategy for learning this distribution (exploring) while, simultaneously, using the agent's accumulated samples to identify the current "best" arm so as to maximize profit (exploiting). Different algorithms employ different strategies to balance exploration and exploitation, but a standard definition for the "best" arm is the one that has the highest payout rate associated with it. We will show that, perhaps surprisingly, the definition of "best" arm is more involved when unobserved confounders are present.

This paper complements the vast literature of MAB that encompasses many variants including adversarial bandits (in which an omnipotent adversary can dynamically shift the reward distributions to thwart the player's best strategies) [BFK10, AS95, BS12] contextual bandits (in which the payout,

and therefore the best choice of action, is a function of one or more observed environmental variables) [LZ08, DHK$^+$11, Sli14], and many different constraints and assumptions over the underlying generative model and payout structure [SBCAY14]. For a recent survey, see [BCB12].

This work addresses the MAB problem when *unobserved confounders* are present (called MABUC, for short), which is arguably the most sensible assumption in real-world, practical applications (obviously weaker than assuming the inexistence of confounders). To support this claim, we should first note that in the experimental design literature, Fisher's very motivation for considering randomizing the treatment assignment was to eliminate the influence of unobserved confounders – factors that simultaneously affect the treatment (or bandit arm) and outcome (or bandit payout), but are not accounted for in the analysis. In reality, the reason for not accounting for such factors explicitly in the analysis is that many of them are unknown a priori by the modeller [Fis51].

The study of unobserved confounders is one of the central themes in the modern literature of causal inference. To appreciate the challenges posed by these confounders, consider the comparison between a randomized clinical trial conducted by the Food and Drug Administration (FDA) versus physicians prescribing drugs in their offices. A key tenet in any FDA trial is the use of randomization for the treatment assignment, which precisely protects against biases that might be introduced by physicians. Specifically, physicians may prescribe Drug A for their wealthier patients who have better nutrition than their less wealthy ones, when unknown to the doctors, the wealthy patients would recover without treatment. On the other hand, physicians may avoid prescribing the expensive Drug A to their less privileged patients, who (again unknown to the doctors) tend to suffer less stable immune systems causing negative reactions to the drug. If a naive estimate of the drug's causal effect is computed based on physicians' data (obtained through random sampling, but not random assignment), the drug would appear more effective than it is in practice – a bias that would otherwise be avoided by random assignment. Confounding biases (of variant magnitude) appear in almost any application in which the goal is to learn policies (instead of statistical associations), and the use of randomization of the treatment assignment is one established tool to combat them [Pea00].

To the best of our knowledge, no method in the bandit literature has studied the issue of unobserved confounding explicitly, in spite of its pervasiveness in real-world applications. Specifically, no MAB technique makes a clear-cut distinction between experimental exploration (through random assignment as required by the FDA) and observational data (as given by random sampling in the doctors' offices). In this paper, we explicitly acknowledge, formalize, and then exploit these different types of data-collection. More specifically, our contributions are as follow:

- We show that the current bandit algorithms implicitly attempt to maximize rewards by estimating the experimental distribution, which does not guarantee an optimal strategy when unobserved confounders are present (Section 2).

- Based on this observation, we translate the MAB problem to causal language, and then suggest a more appropriate metric that bandit players should optimize for when unobserved confounders are present. This leads to a new exploitation principle that can take advantage of data collected under both observational and experimental modes (Section 3).

- We empower Thompson Sampling with this new principle and run extensive simulations. The experiments suggest that the new strategy is stats. efficient and consistent (Sec. 4).

## 2 Challenges due to Unobserved Confounders

In this section, we discuss the mechanics of how the maximization of rewards is treated based on a bandit instance with unobserved confounders. Consider a scenario in which a greedy casino decides to demo two new models of slot machines, say $M_1$ and $M_2$ for simplicity, and wishes to make them as lucrative as possible. As such, they perform a battery of observational studies (using random sampling) to compare various traits of the casino's gamblers to their typical slot machine choices. From these studies, the casino learns that two factors well predict the gambling habits of players when combined (unknown by the players): player inebriation and machine conspicuousness (say, whether or not a machine is blinking). Coding both of these traits as binary variables, we let $B \in \{0, 1\}$ denote whether or not a machine is blinking, and $D \in \{0, 1\}$ denote whether or not the gambler is drunk. As it turns out, a gambler's "natural" choice of machine, $X \in \{M_1, M_2\}$, can be modelled by the structural equation indicating the index of their chosen arm (starting at 0):

$$X \leftarrow f_X(B, D) = (D \wedge \neg B) \vee (\neg D \wedge B) = D \oplus B \tag{1}$$

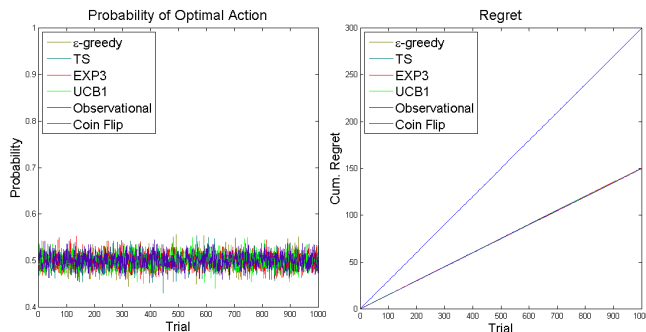

Figure 1: Performance of different bandit strategies in the greedy casino example. Left panel: no algorithm is able to perform better than random guessing. Right panel: Regret grows without bounds.

Moreover, the casino learns that every gambler has an equal chance of being intoxicated and each machine has an equal chance of blinking its lights at a given time, namely, $P(D = 0) = P(D = 1) = 0.5$ and $P(B = 0) = P(B = 1) = 0.5$.

The casino's executives decide to take advantage of these propensities by introducing a new type of reactive slot machine that will tailor payout rates to whether or not it believes (via sensor input, assumed to be perfect for this problem) a gambler is intoxicated. Suppose also that a new gambling law requires that casinos maintain a minimum attainable payout rate for slots of 30%. Cognizant of this new law, while still wanting to maximize profits by exploiting gamblers' natural arm choices, the casino executives modify their new slots with the payout rates depicted in Table 1a.

| (a) | $D = 0$ | | $D = 1$ | |
|---|---|---|---|---|
| | $B = 0$ | $B = 1$ | $B = 0$ | $B = 1$ |
| $X = M_1$ | *0.10 | 0.50 | 0.40 | *0.20 |
| $X = M_2$ | 0.50 | *0.10 | *0.20 | 0.40 |

| (b) | $P(y\|X)$ | $P(y\|do(X))$ |
|---|---|---|
| $X = M_1$ | 0.15 | 0.3 |
| $X = M_2$ | 0.15 | 0.3 |

Table 1: (a) Payout rates decided by reactive slot machines as a function of arm choice, sobriety, and machine conspicuousness. Players' natural arm choices under $D, B$ are indicated by asterisks. (b) Payout rates according to the observational, $P(Y = 1|X)$, and experimental $P(Y = 1|do(X))$, distributions, where $Y = 1$ represents winning (shown in the table), and $0$ otherwise.

The state, blind to the casino's payout strategy, decides to perform a randomized study to verify whether the win rates meet the 30% payout requisite. Wary that the casino might try to inflate payout rates for the inspectors, the state recruits random players from the casino floor, pays them to play a random slot, and then observes the outcome. Their randomized experiment yields a favorable outcome for the casino, with win rates meeting precisely the 30% cutoff. The data looks like Table 1b (third column), assuming binary payout $Y \in \{0, 1\}$, where $0$ represents losing, and $1$ winning.

As students of causal inference and still suspicious of the casino's ethical standards, we decide to go to the casino's floor and observe the win rates of players based on their natural arm choices (through random sampling). We encounter a distribution close to Table 1b (second column), which shows that the casino is actually paying ordinary gamblers only 15% of the time.

In summary, the casino is at the same time (1) exploiting the natural predilections of the gamblers' arm choices as a function of their intoxication and the machine's blinking behavior (based on eq. 1), (2) paying, on average, less than the legally allowed (15% instead of 30%), and (3) fooling state's inspectors since the randomized trial payout meets the 30% legal requirement.

As machine learning researchers, we decide to run a battery of experiments using the standard bandit algorithms (e.g., $\epsilon$-greedy, Thompson Sampling, UCB1, EXP3) to test the new slot machines on the casino floor. We obtain data encoded in Figure 1a, which shows that the probability of choosing the correct action is no better than a random coin flip even after a considerable number of steps. We note, somewhat surprised, that the cumulative regret (Fig. 1b) shows no signs of abating, and

that we are apparently unable to learn a superior arm. We also note that the results obtained by the standard algorithms coincide with the randomized study conducted by the state (purple line).

Under the presence of unobserved confounders such as in the casino example, however, $P(y|do(X))$ does not seem to capture the information required to maximize payout, but rather the average payout akin to choosing arms by a coin flip. Specifically, the payout given by coin flipping is the same for both machines, $P(Y = 1|do(X = M_1)) = P(Y = 1|do(X = M_2)) = 0.3$, which means that the arms are statistically indistinguishable in the limit of large sample size. Further, if we consider using the observational data from watching gamblers on the casino floor (based on their natural predilections), the average payoff will also appear independent of the machine choice, $P(Y = 1|X = M_1) = P(Y = 1|X = M_2) = 0.15$, albeit with an even lower payout. [1]

Based on these observations, we can see why no arm choice is better than the other under either distribution alone, which explains the reason any algorithm based on these distributions will certainly fail to learn an optimal policy. More fundamentally, we should be puzzled by the disagreement between observational and interventional distributions. This residual difference may be encoding knowledge about the unobserved confounders, which may lead to some indication on how to differentiate the arms. This indeed may lead to some indication on how to differentiate the arms as well as a sensible strategy to play better than pure chance. In the next section, we will use causal machinery to realize this idea.

## 3 Bandits as a Causal Inference Problem

We will use the language of *structural causal models* [Pea00, Ch. 7] for expressing the bandit data-generating process and for allowing the explicit manipulation of some key concepts in our analysis – i.e., confounding, observational and experimental distributions, and counterfactuals (to be defined).

**Definition 3.1. (Structural Causal Model)** ([Pea00, Ch. 7]) A structural causal model $M$ is a 4-tuple $\langle U, V, f, P(u) \rangle$ where:

1. $U$ is a set of background variables (also called exogenous), that are determined by factors outside of the model,

2. $V$ is a set $\{V_1, V_2, ..., V_n\}$ of observable variables (also called endogenous), that are determined by variables in the model (i.e., determined by variables in $U \cup V$),

3. $F$ is a set of functions $\{f_1, f_2, ..., f_n\}$ such that each $f_i$ is a mapping from the respective domains of $U_i \cup PA_i$ to $V_i$, where $U_i \subseteq U$ and $PA_i \subseteq V \setminus V_i$ and the entire set $F$ forms a mapping from $U$ to $V$. In other words, each $f_i$ in $v_i \leftarrow f_i(pa_i, u_i), i = 1, ..., n$, assigns a value to $V_i$ that depends on the values of the select set of variables $(U_i \cup PA_i)$, and

4. $P(u)$ is a probability distribution over the exogenous variables.

Each structural model $M$ is associated with a directed acyclic graph $G$, where nodes correspond to endogenous variables $V$ and edges represent functional relationships – i.e., there exists an edge from $X$ to $Y$ whenever $X$ appears in the argument of $Y$'s function. We define next the MABUC problem within the structural semantics.

**Definition 3.2. (K-Armed Bandits with Unobserved Confounders)** A K-Armed bandit problem with unobserved confounders is defined as a model $M$ with a reward distribution over $P(u)$ where:

1. $X_t \in \{x_1, ..., x_k\}$ is an observable variable encoding player's arm choice from one of $k$ arms, decided by Nature in the observational case, and $do(X_t = \pi(x_0, y_0, ..., x_{t-1}, y_{t-1}))$, for strategy $\pi$ in the experimental case (i.e., when the strategy decides the choice),

2. $U_t$ represents the unobserved variable encoding the payout rate of arm $x_t$ as well as the propensity to choose $x_t$, and

3. $Y_t \in 0, 1$ is a reward (0 for losing, 1 for winning) from choosing arm $x_t$ under unobserved confounder state $u_t$ decided by $y_t = f_y(x_t, u_t)$.

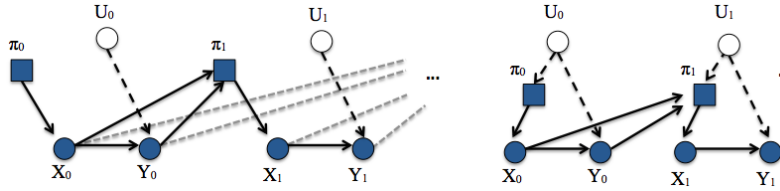

Figure 2: (a) Model for the standard MAB sequential decision game. (b) Model for the MABUC sequential decision game. In each model, solid nodes denote observed variables and open nodes represent unobserved variables. Square nodes denote the players strategys arm choice at time $t$. Dashed lines illustrate influences on future time trials that are not pictured.

First note that this definition also applies to the MAB problem (without confounding) as shown in Fig. 2a. This standard MAB instance is defined by constraining the MABUC definition such that $U_t$ affects only the outcome variable $Y_t$ – there is no edge from $U_t$ to $X_t$ (Def. 3.2.2). In the unconfounded case, it is clear that $P(y|do(x)) = P(y|x)$ [Pea00, Ch. 3], which means that that payouts associated with flipping a coin to randomize the treatment or observing (through random sampling) the player gambling on the casino's floor based on their natural predilections will yield the same answer. The variable $U$ carries the unobserved payout parameters of each arm, which is usually the target of analysis. [2] [3]

Fig. 2b provides a graphical representation of the MABUC problem. Note that $\pi_t$ represents the system's choice policy, which is affected by the unobserved factors encoded through the arrow from $U_t$ to $\pi_t$. One way to understand this arrow is through the idea of players' natural predilections. In the example from the previous section, the predilection would correspond to the choices arising when the gambler is allowed to play freely on the casino's floor (e.g., drunk players desiring to play on the blinking machines) or doctors prescribing drugs based on their gut feeling (e.g., physicians prescribing the more expensive drug to their wealthier patients). These predilections are encoded in the observational distribution $P(y|x)$. On the other hand, the experimental distribution $P(y|do(x))$ encodes the process in which the natural predilections are overridden, or ceased by external policies. In our example, this distribution arises when the government's inspectors flip a coin and send gamblers to machines based on the coin's outcome, regardless of their predilections.

Remarkably, it is possible to use the information embedded in these distinct data-collection modes (and their corresponding distributions) to understand players' predilections and perform better than random guessing in these bandit instances. To witness, assume there exists an oracle on the casino's floor operating by the following protocol. The oracle observes the gamblers until they are about to play a given machine. The oracle intercepts each gambler who is about to pull the arm of machine $M_1$, for example, and suggests the player to contemplate whether following his predilection ($M_1$) or going against it (playing $M_2$) would lead to a better outcome. The drunk gambler, who is a clever machine learning student and familiar with Fig. 1, says that this evaluation cannot be computed a priori. He affirms that, despite spending hours on the casino estimating the payoff distribution based on players' natural predilections (namely, $P(y|x)$), it is not feasible to relate this distribution with the hypothetical construction *what would have happened had he decided to play differently*. He also acknowledges that the experimental distribution $P(y|do(x))$, devoid of the gamblers' predilections, does not support any clear comparison against his personal strategy. The oracle says that this type of reasoning is possible, but first one needs to define the concept of counterfactual.

**Definition 3.3. (Counterfactual)** ([Pea00, pp. 204]) Let $X$ and $Y$ be two subsets of exogenous variables in $V$. The counterfactual sentence "$Y$ would be $y$ (in situation $u$), had $X$ been $x$" is interpreted as the equality with $Y_x(u) = y$, with $Y_x(u)$ being the potential response of $Y$ to $X = x$.

This definition naturally leads to the judgement suggested by the oracle, namely, "would I (the agent) win ($Y = 1$) had I played on machine $M_1$ ($X = 1$)", which can be formally written as $Y_{X=1} = 1$ (we drop the $M$ for simplicity). Assuming that the agent's natural predilection is to play machine 1, the oracle suggests an introspection comparing the odds of winning following his intuition or going against it. The former statement can be written in counterfactual notation, probabilistically, as $E(Y_{X=1} = 1|X = 1)$, which reads as "the expected value of winning ($Y = 1$) had I play machine 1 given that I am about to play machine 1", which contrasts with the alternative hypothesis $E(Y_{X=0} = 1|X = 1)$, which reads as "the expected value of winning ($Y = 1$) had I play machine 1 given that I am about to play machine 0". This is also known in the literature as the *effect of the treatment on the treated* (ETT) [Pea00]. So, instead of using a decision rule comparing the average payouts across arms, namely (for action $a$),

$$\operatorname*{argmax}_a \ E(Y|do(X = a)), \tag{2}$$

which was shown in the previous section to be insufficient to handle the MABUC, we should consider the rule using the comparison between the average payouts obtained by players for choosing in favour or against their intuition, respectively,

$$\operatorname*{argmax}_a \ E(Y_{X=a} = 1|X = x), \tag{3}$$

where $x$ is the player's natural predilection and $a$ is their final decision. We will call this procedure RDC (regret decision criterion), to emphasize the counterfactual nature of this reasoning step and the idea of following or disobeying the agent's intuition, which is motivated by the notion of regret. Remarkably, RDC accounts for the agents individuality and the fact that their natural inclination encodes valuable information about the confounders that also affect the payout. In the binary case, for example, assuming that $X = 1$ is the player's natural choice at some time step, if $E(Y_{X=0} = 1|X = 1)$ is greater than $E(Y_{X=1} = 1|X = 1)$, this would imply that the player should refrain of playing machine $X = 1$ to play machine $X = 0$.

Assuming one wants to implement an algorithm based on RDC, the natural question that arises is how the quantities entailed by Eq. 3 can be computed from data. For the factors in the form $E(Y_{X=1} = 1|X = 1)$, the consistency axiom [Pea00, pp. 229] implies that $E(Y_{X=1} = 1|X = 1) = E(Y = 1|X = 1)$, where the l.h.s. is estimable from observational data. Counterfactuals in the form $E(Y_{X=a} = 1|X = x)$, where $a \neq x$, can be computed in the binary case through algebraic means [Pea00, pp. 396-7]. For the general case, however, ETT is not computable without knowledge of the causal graph. [4] Here, ETT will be computed in an alternative fashion, based on the idea of intention-specific randomization. The main idea is to randomize intention-specific groups, namely, interrupt any reasoning agent before they execute their choice, treat this choice as intention, delibarte, and then act. We discuss next about the algorithmic implementation of this randomization.

## 4 Applications & Experiments

Based on the previous discussion, we can revisit the greedy casino example from Section 2, apply RDC and use the following inequality to guide agent's decisions:

$$E(Y_{X=0}|X = 1) > E(Y_{X=1}|X = 1) \Leftrightarrow E(Y_{X=0}|X = 1) > P(Y|X = 1) \tag{4}$$

There are different ways of incorporating this heuristic into traditional bandit algorithms, and we describe one such approach taking the Thompson Sampling algorithm as the basis [OB10, CL11, AG11]. (For simulation source code, see https://github.com/ucla-csl/mabuc )

Our proposed algorithm, Causal Thompson Sampling ($TS^C$) takes the following steps: (1) $TS^C$ first accepts an observational distribution as input, which it then uses to seed estimates of ETT quantities; i.e., for actions $a$ and intuition $x$, by consistency we may seed knowledge of $E(Y_{X=a}|X = x) = P_{obs}(y|x), \forall a = x$. With large samples from an input set of observations, this seeding reduces (and possibly eliminates) the need to explore the payout rates associated with following intuition, leaving only the "disobeying intuition" payout rates left for the agent to learn. As such, (2) at each time step, our oracle observes the agent's arm-choice predilection, and then uses RDC to deter-

mine their best choice.[5] Lastly, note that our seeding in (2) immediately improves the accuracy of our comparison between arms, viz. that a superior arm will emerge more quickly than had we not seeded. We can exploit this early lead in accuracy by weighting the more favorable arm, making it more likely to be chosen earlier in the learning process (which empirically improves the convergence rate as shown in the simulations).

---

**Algorithm 1** Causal Thompson Sampling $(TS^C)$

---

1: **procedure** $\text{TS}^C(P_{obs}, \text{T})$
2:      $E(Y_{X=a}|X) \leftarrow P_{obs}(y|X)$          (seed distribution)
3:      **for** $t = [1, ..., T]$ **do**
4:          $x \leftarrow intuition(t)$      (get intuition for trial)
5:          $Q_1 \leftarrow E(Y_{X=x'}|X=x)$      (estimated payout for counter-intuition)
6:          $Q_2 \leftarrow P(y|X=x)$      (estimated payout for intuition)
7:          $w \leftarrow [1, 1]$      (initialize weights)
8:          $bias \leftarrow 1 - |Q_1 - Q_2|$      (compute weighting strength)
9:          **if** $Q_1 > Q_2$ **then** $w[x] \leftarrow bias$ **else** $w[x'] \leftarrow bias$      (choose arm to bias)
10:        $a \leftarrow max(\beta(s_{M_1,x}, f_{M_1,x}) \times w[1], \beta(s_{M_2,x}, f_{M_2,x}) \times w[2])$      (choose arm) [6]
11:        $y \leftarrow pull(a)$      (receive reward)
12:        $E(Y_{X=a}|X=x) \leftarrow y|a, x$      (update)

---

In the next section, we provide simulations to support the efficacy of $TS^C$ in the MABUC context. For simplicity, we present two simulation results for the model described in Section 2.[7] Experiment 1 employs the "Greedy Casino" parameterization found in Table 1, whereas Experiment 2 employs the "Paradoxical Switching" parameterization found in Table 2. Each experiment compares the performance of traditional Thompson Sampling bandit players versus $TS^C$.

| (a) | $D = 0$ | | $D = 1$ | |
|---|---|---|---|---|
| | $B = 0$ | $B = 1$ | $B = 0$ | $B = 1$ |
| $X = M_1$ | *0.40 | 0.30 | 0.30 | *0.40 |
| $X = M_2$ | 0.60 | *0.10 | *0.20 | 0.60 |

| (b) | $P(y|X)$ | $P(y|do(X))$ |
|---|---|---|
| $X = M_1$ | 0.4 | 0.35 |
| $X = M_2$ | 0.15 | 0.375 |

Table 2: "Paradoxical Switching" parameterization. (a) Payout rates decided by reactive slot machines as a function of arm choice, sobriety, and machine conspicuousness. Players' natural arm choices under $D, B$ are indicated by asterisks. (b) Payout rates associated with the observational and experimental distributions, respectively.

**Procedure.** All reported simulations are partitioned into rounds of $T = 1000$ trials averaged over $N = 1000$ Monte Carlo repetitions. At each time step in a single round, (1) values for the unobserved confounders $(B, D)$ and observed intuitive arm choice $(x)$ are selected by their respective structural equations (see Section 2), (2) the player observes the value of $x$, (3) the player chooses an arm based on their given strategy to maximize reward (which may or may not employ $x$), and finally, (4) the player receives a Bernoulli reward $Y \in \{0, 1\}$ and records the outcome.

Furthermore, at the start of every round, players possess knowledge of the problem's observational distribution, i.e., each player begins knowing $P(Y|X)$ (see Table 2b). However, only causally-empowered strategies will be able to make use of this knowledge, since this distribution is not, as we've seen, the correct one to maximize.

**Candidate algorithms.** *Standard Thompson Sampling* $(TS)$ attempts to maximize rewards based on $P(y|do(X))$, ignoring the intuition $x$. *Z-Empowered Thompson Sampling* $(TS^Z)$ treats the

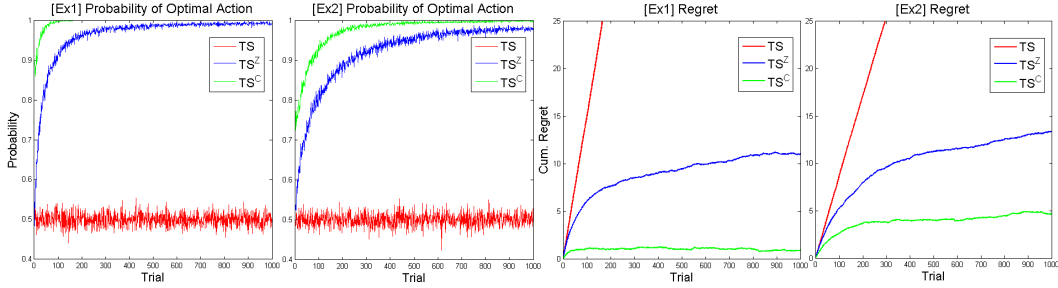

Figure 3: Simulation results for Experiments 1 and 2 comparing standard Thompson Sampling ($TS$), Z-Empowered Thompson Sampling ($TS^Z$), and Causal Thompson Sampling ($TS^*$)

predilection as a new context variable, $Z$, and attempts to maximize based on $P(y|do(X), Z)$ at each round. *Causal Thompson Sampling* ($TS^C$), as described above, employs the ETT inequality and input observational distribution.

**Evaluation metrics.** We assessed each algorithms' performances with standard bandit evaluation metrics: (1) the probability of choosing the optimal arm and (2) cumulative regret. As in traditional bandit problems, these measures are recorded as a function of the time step $t$ averaged over all $N$ round repetitions. Note, however, that traditional definitions of regret are not phrased in terms of unobserved confounders; our metrics, by contrast, compare each algorithm's chosen arm to the optimal arm for a given instantiation of $B_t$ and $D_t$, even though these instantiations are never directly available to the players. We believe that this is a fair operationalization for our evaluation metrics because it allows us to compare regret experienced by our algorithms to a truly optimal (albeit hypothetical) policy that has access to the unobserved confounders.

**Experiment 1: "Greedy Casino."** The Greedy Casino parameterization (specified in Table 1) illustrates the scenario where each arm's payout appears to be equivalent under the observational and experimental distributions alone. Only when we concert the two distributions and condition on a player's predilection can we obtain the optimal policy. Simulations for Experiment 1 support the efficacy of the causal approach (see Figure 3). Analyses revealed a significant difference in the regret experienced by $TS^Z$ ($M = 11.03, SD = 15.96$) compared to $TS^C$ ($M = 0.94, SD = 15.39$), $t(999) = 14.52, p < .001$. Standard $TS$ was, predictably, not a competitor experiencing high regret ($M = 150.47, SD = 14.09$).

**Experiment 2: "Paradoxical Switching."** The Paradoxical Switching parameterization (specified in Table 2a) illustrates the scenario where one arm ($M_1$) appears superior in the observational distribution, but the other arm ($M_2$) appears superior in the experimental. Again, we must use causal analyses to resolve this ambiguity and obtain the optimal policy. Simulations for Experiment 2 also support the efficacy of the causal approach (see Figure 3). Analyses revealed a significant difference in the regret experienced by $TS^Z$ ($M = 13.39, SD = 17.15$) compared to $TS^C$ ($M = 4.71, SD = 17.90$), $t(999) = 11.28, p < .001$. Standard $TS$ was, again predictably, not a competitor experiencing high regret ($M = 83.56, SD = 15.75$).

# 5 Conclusions

In this paper, we considered a new class of bandit problems with unobserved confounders (MABUC) that are arguably more realistic than traditional formulations. We showed that MABUC instances are not amenable to standard algorithms that rely solely on the experimental distribution. More fundamentally, this lead to an understanding that in MABUC instances the optimization task is not attainable through the estimation of the experimental distribution, but relies on both experimental and observational quantities rooted in counterfactual theory and based on the agents' predilections. To take advantage of our findings, we empowered the Thompson Sampling algorithm in two different ways. We first added a new rule capable of improving the efficacy of which arm to explore. We then jumpstarted the algorithm by leveraging non-experimental (observational) data that is often available, but overlooked. Simulations demonstrated that in general settings these changes lead to a more effective decision-making with faster convergence and lower regret.

## Footnotes

\*The authors contributed equally to this paper.

[1]One may surmise that these ties are just contrived examples, or perhaps numerical coincidences, which do not appear in realistic bandit instances. Unfortunately, that's not the case as shown in the other scenarios discussed in the paper. This phenomenon is indeed a manifestation of the deeper problem arising due to the lack of control for the unobserved confounders.

[2] On a more fundamental level, it is clear that unconfoundedness is (implicitly) assumed not to hold in the general case. Otherwise, the equality between observational and experimental distributions would imply that no randomization of the action needs to be carried out since standard random sampling would recover the same distribution. In this case, this would imply that many works in the literature are acting in a suboptimal way since, in general, experiments are more expensive to perform than collecting data through random sampling.

[3] The interventional nature of the MAB problem is virtually not discussed in the literature, one of the few exceptions is the causal interpretation of Thompson Sampling established in [OB10].

[4]Graphical conditions for identifying ETT [Pea00, SP07] are orthogonal to the bandit problem studied in this paper, since no detailed knowledge about the causal graph (as well as infinite samples) is assumed here.

[5]Note that using predilection as a criteria for the inequality does not uniquely map to the contextual bandit problem. To understand this point, note that not all variables are equally legitimate for confounding control in causal settings, while the agent's predilection is certainly one of such variables in our setup. Specifically when considering whether a variable qualifies as a causal context requires a much deeper understanding of the data generating model, which is usually not available in the general case.

[6]The notation: $\beta(s_{M_k,x}, f_{M_k,x})$ means to sample from a Beta distribution with parameters equal to the successes encountered choosing action $x$ on machine $M_k(s_{M_k,x})$ and the failures encountered choosing action $x$ on that machine $(f_{M_k,x})$.

[7]For additional experimental results and parameterizations, see Appendix [BFP15].

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
