[Reviews · NeurIPS 2015]

Submitted by Assigned_Reviewer_1

The paper "Bandits with unobs. confounders: a causal approach" addresses the problem of bandit learning. It is assumed that in the observational setting, the player's decision is influenced by some unobserved context. If we randomize the player's decision, however, this intention is lost. The key idea is now that, using the available data from both scenarios, one can infer whether one should overrule the player's intention. Ultimately, this leads to the following strategy: observe the player's intention and then decide whether he should act accordingly or pull the other arm.

I like the problem the paper addresses and the way it approaches it. Although I would have liked to do so, however, I cannot strongly recommend the acceptance of this paper due to

1) it contains almost no material beyond an idea 2) limitations to bivariate variables? 3) method is not properly compared to contextual bandits, experiments available only on two artificial toy data sets

I definitely encourage the authors to continue this line of research and provide a more thorough discussion of this problem.

- the key idea seems to be (7), which computes a counterfactual statement from observational and interventional data. unfortunately, this is already known (e.g. "What Counterfactuals Can Be Tested" Shpitser, Pearl, p.353).

- also, i am not sure how the idea generalizes to variables with more than 2 states (i.e., #arms and/or pay-off). it is known that in general, counterfactuals cannot be inferred from obs. and int. data. This seems to be a major limitation that is not discussed properly.

- the assumption that the intention of the player can be measured before he pulls an arm is quite strong. but if it can, this problem is "just" a contextual bandit (e.g. use TS^Z). It remained unclear to me, why in the simulations, the proposed method got access to the observational data, while the TS^Z approach did not (if it did, it should be at least as good!?). I think, one could rewrite the whole motivation of the paper: consider a situation, in which one first observes a _different_ player (obs. data) playing a bandit and then oneself sits down in order to gather the interventional data. this requires the assumption that all players develop the same intention. but then it seems to boil down to a contextual bandit with a context that is partially hidden for some data points. Is there any literature on this problem?

- the first concrete idea (as far as I can tell) is in the last paragraph on page 6 (for a conference paper, the introduction is too long)

- there is no code available

- there are only toy examples

- in many places the paper is badly written/unclear. i only give few examples, there are more * bottom page 2, there are two machines (sometimes two arms), but D indicates whether "the" machine is blinking * Y = {0,1} doesn't make sense if Y is a rand.var. * the authors confuse E and P many times. E(Y) = P(Y=1) if Y takes values in {0,1} and change notation. * alg. 1 contains many unclear statements, e.g. beta and f are not defined properly, what is x' etc. * the writing f(x,a) for all a = x, is not very sensible, bottom p. 6 * the abstract is a bit unclear "passive data through random sampling"

- maybe focus more on the example?

- the colors in fig. 1 are indistinguishable

minor - greendy casino - use \eqref - shouldn't (8) be <=> ? - is great than - citations are very inhomogeneous - random experiments do not control for but eliminate the influence of hidden confounders. - what are the grey lines in Fig. 2? - I would not regard Def. 3.3. as a definition
Summary: neat idea but too little content: no novel theory, no real experiments, not clear (to me) whether this approach generalizes to non-bivariate variables

Submitted by Assigned_Reviewer_2

This paper is concerned with the multi-armed bandit (MAB) problem when unobserved confounders exist. The author showed that the current MAB algorithms actually attempt to maximize rewards according to the experimental distribution, which is not optimal in the confounding case, and proposed to make use of the effect of the treatment on the treated (ETT), i.e., by comparing the average payouts obtained by players for going in favor of or against their intuition. To me, the paper is interesting because it addresses the confounding issue in MAB and proposed a way to estimate some properties of the confounder (related to the casino's payout strategy in the given example) based on ETT.

Without a causal thinking, it would be difficult to see the effect of confounders in the studied problem and handle it effectively.

I have the following suggestions. 1. It is clear that the causal thinking helps formulate the potential problem MAB and tackle the problem properly, but it would be more helpful to clearly explain what the authors mean by "causal approach." More specifically, what is the difference between the causal knowledge and a graphical model? What is the unique advantage of using causal knowledge? To me, standard Thompson Sampling is also a causal method since it aims to maximize P(y|do(X)).

2. According to my understanding, by using ETT, the authors actually tried to infer some properties of the confounders. If this is the case, it would be nice to make this more direct and discuss the possibility of finding alternative approaches. 3. I am wondering if the idea can be explained more concisely and clearly with the help of some simple graphical representations.

Minor points: - [LZ08, Sli14, Sli14]. - On page 3, should P(y|X) be P(y=1|X)? - this criteria. - We then jump started the algorithm by...
Summary: This paper is concerned with the multi-armed bandit (MAB) problem when unobserved confounders exist, and the proposed approach was inspired by a causal treatment. It is interesting because it addresses the confounding issue in MAB and proposed a way to estimate some properties of the confounder (related to the casino's payout strategy in the given example) based on the effect of the treatment on the treated.

Submitted by Assigned_Reviewer_3

This paper describes a new variation on multi-arm bandit problems where the agent's actions are confounded in a particular way. It describes a way to modify Thompson sampling so that performs better on at least some of these problems than traditional Thompson sampling.

The paper addresses an interesting phenomenon and is clearly written for the most part, but it took me a little while to understand the way the confounders worked in the casino example in the paper. At first glance, one might think that the blinking light on the slot machines (B) and the drunkenness of the patron (D) could be either modified or observed in lines 153-159, where we read about a hypothetical attempt to optimize reward using traditional Thompson sampling. If those factors were observable or subject to intervention -- and I'd think they would be, in reality -- then it would be straightforward to do better than the 30% reward rate that's given. The paper eventually makes it clear that both of these variables are unobserved and unalterable. It would help if this were explicit early in the example, or if the cover story were modified to make this aspect more intuitive.

More importantly, I worry about the generality of the algorithm, for the following reasons:

1. The paper focuses on the 2-arm bandit case. How does the algorithm work in the presence of more arms?

2. The paper considers two examples in which the payoff structure is neatly matched to the algorithm. One would do well in the greedy casino case by replacing the nominal arms with two new ones: the intuition arm and the counter-intuition arm, and quickly learn to pull the latter. How well does the algorithm perform if the intuition is independent of the payoff structure, e.g., if we take a classical MAB problem and include intuition as an irrelevant random variable. The examples in the supplementary material help alleviate this concern, but I'm not totally convinced that one should simply throw out the observational data in the traditional Thompson sampling case; would doing so significantly hurt that method in the actually-confounded examples? In the unconfounded case, using the observational data (rightly or wrongly), it appears that the expected TS^C payoff in this case would be at least slightly worse. It's much easier to get excited about causal Thompson sampling if it can be usefully applied to cases where there just might be an important, and exploitable, unobserved confounder.

A minor note: it may be helpful to do another round of proofreading. The paper has several typos, including writing "Thompson" as "Thomson".
Summary: Contains an interesting idea and an algorithm that performs well in some potentially-important circumstances. However, the case for the scope and significance of the method could be stronger.

Submitted by Assigned_Reviewer_4

The paper discusses the multi-arm bandit problem in models with unobserved confounders. The presence of unobserved confounders implies that it is necessary to distinguish observational and interventional distributions. The authors use counterfactuals for the distinction and discuss the implications for bandit algorithms that try to maximise reward. They describe a causal version of the Thomson sampling algorithm.

There have been other attempts at linking optimal descision making with unobserved confounders in the statistics literature, e.g. Dawid and Didelez (statistics Surveys 2010) uses a decision theoretic framework and Murphy (2003, JRSSB) uses counterfactuals. Clearly the focus of the other work is not on bandit algorithms but it would be helpful to comment on any similarities.

The definition of the quantities in the definition of ETT is not very intuitive. For example E(Y_{X=0} = 1| X = 1) should be interpreted along the lines of "expected value of Y if I had played machine 0, given that I actually played machine 1".

The quantity ETT is actually the contrast between E(Y_{X=1} = 1| X = 1) and E(Y_{X=0} = 1| X = 1). It seems to refer to one of the quantities in the paper.

Why is the focus on ETT? The usual justification is that the treated population is the population of interest or the agent of interest is those that naturally choose machine 1. In this case, the focus could have been on agents that choose machine 0 as well.

In the proof of (6), give an intuitive explanation of the consistency axiom.

Near the bottom of page 6, what are the "disobeying intuition" payout rates?
Summary: The section of the paper which discusses the causal implications of unobserved confounders is not novel. It describes standard results in the causal inference literature about identifying interventions from observational distributions. However the introduction of unobserved confounders to the bandit algorithms is a useful extension and seems to be new. This is an important bridge between the causal inference literature and the literature on bandit problems.

Author Feedback
Author rebuttal: We appreciate the reviewers' feedback, suggestions, and general encouragement to pursue this line of research. Due to space constraints, we will answer the most pressing concerns that were raised by the reviewers.

* Re:  "boils down to contextual bandit problem" / "no material beyond an idea"
We agree that, viewed as a "conditioning on predilection" problem, contextual bandit solvers can successfully navigate the MABUC problem by treating "intent" as a context -- this is, indeed the essence of what we are proposing: an agent observes its intention, pauses to deliberate, and then acts to maximize reward, conditioned on that intention as evidence.

Note that the prospect of an agent knowing its own intention is not a strong assumption in many settings; simply interrupt any reasoning agent before they execute their choice, treat this choice as intention, deliberate, and then act. Moreover, and this is one advantage of an ETT reasoner over a contextual reasoner, the former need not record its intentions -- the bare actions suffice (in the binary case).

With that said, our paper's contribution is threefold: (1) We emphasize and formalize the distinction between action intended and action executed in the MAB setting; this has not previously been considered in the MAB literature. (2) Using this distinction, we are able to properly incorporate observational data sets to "jump start" bandit algorithms with access to such cheap data. (3) We show that this seeding strategy together with the maximization of the ETT function (instead of experimental distribution as commonly believed) can be incorporated in the Thompson Sampling algorithm and lead to significant improvement of its performance. 

We therefore contend that rating our paper as containing "no material beyond an idea" is perhaps too strong; we believe that all known bandit algorithms should be enhanced with the considerations raised by our paper as exemplified in our algorithm.

* Re: limitations to binary action. Quoting Reviewer 2:
"...the key idea seems to be (7), ...  this is already known (e.g. "What Counterfactuals Can Be Tested" Shpitser, Pearl, p.353)."

We should certainly cite Shpritser-Pearl's (SP) paper, where the binary case and equation (7) are acknowledged as an "algebraic trick." However, the novelty of our paper lies not in deriving Eq. (7), but in calling attention to the fact the ETT can serve as a guiding principle for decisions in sequential settings, and more importantly, be computed in an alternative fashion.

Note that, while the identification conditions of SP require a causal graph for their validity (and infinite samples), our approach is entirely data driven; it requires no modeling assumptions whatsoever. This essential difference is what makes the ETT strategy applicable to the MABUC problem, this is also the reason why we do not need graphs to justify the ETT strategy.

(Note also that Pearl moved from labeling (7) an "algebraic trick" to considering it a "triumph of counterfactual analysis" (Causality 2009, pages 396-7), and later turned it into a "paradox of inevitable regret" (Pearl, JCI, 2013). )

Another advance embedded in our algorithm is the idea of intention-specific randomization, which opens a path to estimating counterfactuals beyond the binary treatment, which is one of the limitations of other approaches.

* Re: toy data sets and code
The submission's accompanying appendix contains additional data sets illustrating the algorithm's efficacy across a wide range of parameterizations. Although these are "toy" data sets, they represent models for which the current decision literature would be powerless to navigate without distinguishing intention from execution. Moreover, they were crafted to capture qualitatively different relationships across observational, interventional, and counterfactual distributions.

Simulation code will be available as soon as the paper is published.

* Re: interpretation of ETT
The traditional ETT definition of E(Y_{X=0} = 1 | X = 1) is interpreted as "the expected value of Y if I had played machine 0, given that I actually played machine 1," with the observation occurring in the past tense. However, in the context of decision making, X = 1 stands for the player's intent because it is driven by the same factors that led to that action without deliberation.

* Re: Clarifying the consistency axiom
In the "possible world" semantics, consistency represents the idea that our world is closest to itself than any other possible world. In empirical context, it says that if you take a drug and recover then you would also recover if an experimenter assigned you to take that drug. Consistency is a theorem in counterfactual logic, but is considered an "assumption" by some practitioners.